# Application of the Theory of Planned Behavior in Autonomous Vehicle-Pedestrian Interaction

Farrukh Hafeez [1,*], Usman Ullah Sheikh [1], Abdullahi Abubakar Mas'ud [2], Saud Al-Shammari [2], Muhammad Hamid [3] and Ameer Azhar [2]

1    School of Electrical Engineering, Universiti Teknologi Malaysia, Johor Bahru 81310, Malaysia; usman@fke.utm.my
2    Electrical and Electronic Engineering Technology Department, Jubail Industrial College, Al Jubail 35718, Saudi Arabia; masud_a@jic.edu.sa (A.A.M.); alshammari_s@jic.edu.sa (S.A.-S.); azhar_a@jic.edu.sa (A.A.)
3    Engineering Technology Department, Community College of Qatar, Doha P.O. Box 7344, Qatar; muhammad.hamid@ccq.edu.qa
*    Correspondence: farrukhhafeez82@hotmail.com

**Abstract:** Automobile manufacturers, alongside technology providers, researchers, and public agencies, are conducting extensive testing to design autonomous vehicles (AVs) algorithms that will provide a complete understanding of road users, specifically pedestrians. Pedestrian behavior and actions determination are highly unpredictable depending on behavioral beliefs, context, and socio-demographic variables. Context includes everything that potentially affects one's behavior; in AVs–pedestrian interaction, context may consist of weather conditions, road structure, social factors norms, and traffic volume. These influencing elements, therefore, need to be focused on during the development of pedestrian interaction algorithms. For this purpose, the pedestrian behavior questionnaire for FAVs (PBQF) is designed based on the theory of planned behavior (TPB). A total of almost 1000 voluntary participants completed this multilingual survey. As socio-demographic values and physiological perception varies with local norms, regions, and ethnicity, participants from 27 countries were therefore chosen to account for this variation. One of the key findings of this study is the influence of pedestrian attributes and the context on pedestrian behavior. Pedestrian action cannot be understood without visual observation of the pedestrian themselves and their context. The findings showed that pedestrians build communication with vehicles based on their driving styles. The vehicle's driving style leads pedestrians to think that the vehicle is human-driven or autonomous. The results also revealed that pedestrians use several cues to show their intention. The general perception of AVs was also analyzed, and the communication between AVs and pedestrians with different displaying options was investigated.

**Keywords:** autonomous vehicle; cross-country comparison; pedestrian context; pedestrian behavior; awareness; adoption; communication mode

## 1. Introduction

Transportation is one of humanity's most essential needs, and it has evolved with the passage of time. Automobile manufacturers, researchers, and government officials have all been working on and allocating resources to this new technology. More than fifty companies in the United States have been granted permits to test their AV in California [1], with six autonomous driving companies, Cruise, Waymo, Nuro, Zoox, AutoX, and Baidu, currently in the testing phase. However, many enormous challenges remain to be overcome in terms of general perception, policies, and traffic management prior to the deployment of AVs. As a result, automobile manufacturers have made no plans to commercialize AVs. Environmental perception is one of the critical challenges from a technical standpoint [2–5].

The perception process in AVs refers to the vehicle-mounted sensor's understanding of the environment surrounding the vehicle. Curbs, road lanes, obstacles, drivable areas, surrounding vehicles, infrastructure, traffic signs, traffic signals, and road users are all part of environment perception Pedestrians are one of the most vulnerable road users when compared to other road users. A study conducted by Kaur [6] identified key factors influencing the adoption of driverless cars; pedestrian interaction was identified as a key factor in AV adoption and public trust. AVs have to prove that they are safer than the average human-driven car and have a lower accident rate.

According to a World Health Organization report, approximately 1.35 million people are killed in traffic accidents each year, and 50% of those are vulnerable road users. Further investigation into these accidents reveals that 93% of these fatalities occurred in low and middle-income countries, owing to a number of persistent flaws such as unstructured infrastructure, a lack of vehicle maintenance, and violations of traffic rules [7]. To ensure that AVs are fully robust and capable of acting in any unforeseen circumstances, technologists and automobile manufacturers must ensure that on-road AVs are fully robust and capable of acting in any unforeseen circumstances. The society of automotive engineers (SAE) defined a taxonomy of automation levels with required automation targets, beginning with Level 0 as no automation and ending with Level 5 as full automation [8].

A variety of models, ranging from pedestrian detection to intention estimation and future movement prediction, must collaborate for the full and safe interaction of AVs and pedestrians. In order to achieve this level of maturity, pedestrian model levels are mapped with the level of automation defined by the SAE, ranging from simple driver assistance tools to full driving [9]. The requirements for pedestrian models increase with each level, with the initial model requiring detection, the higher model requiring pedestrian recognition and tracking, and the full interaction model requiring psychological and social understanding to interact in any situation. Figure 1 depicts the SAE levels and pedestrian model requirements.

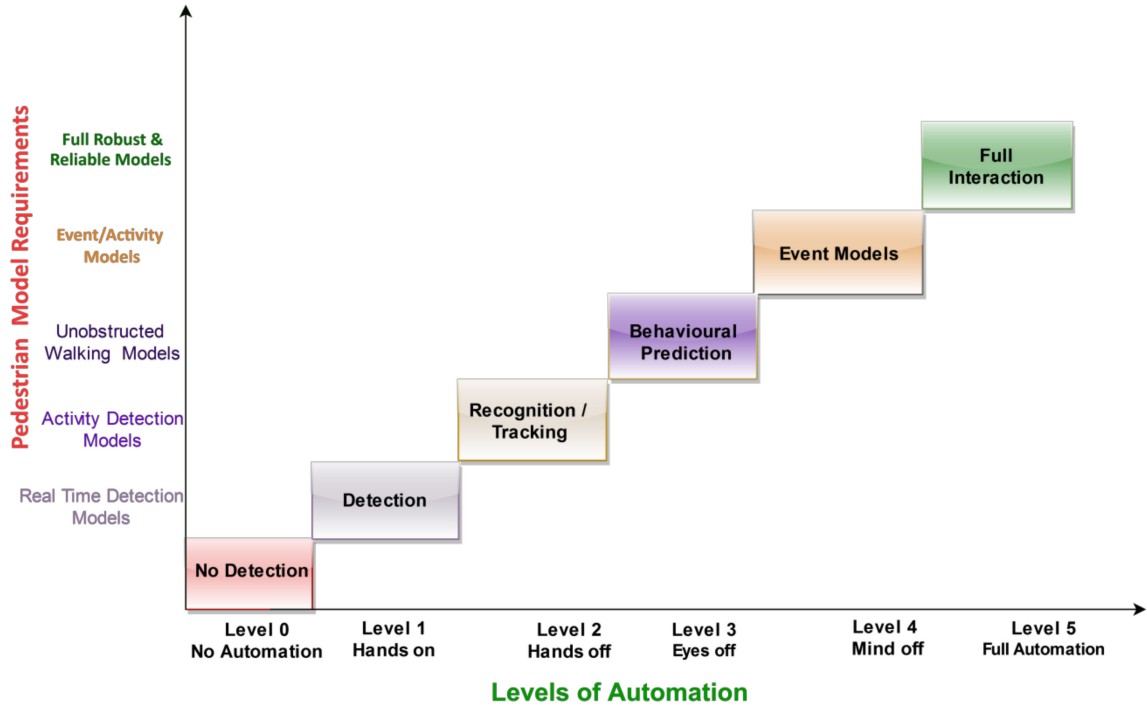

**Figure 1.** Levels of an AV vs. Pedestrian Model Requirement.

A pedestrian behavioral model is a level 3 of automation [10]. This means that the pedestrian model must take into account factors that influence pedestrian behavior and

may influence pedestrian intention and action. Incorporating and considering such factors during the design and implementation phases of algorithms is a critical requirement of behavioral models. Most early research focused on AV technology trust and adoption, communication modalities, and general AV safety perception [11,12], but very few research efforts have been made to investigate pedestrian behavior in the context of AV interaction. Based on human driver perceptions of pedestrians and public perceptions of AVs, this cross-country survey is designed to elucidate behavioral analysis of pedestrians around vehicles. Table 1 lists the various research questions that must be answered, as well as their corresponding motivations. This study was carried out to find answers to these questions.

**Table 1.** Research questions and motivations.

| Questions | Motivations |
| --- | --- |
| Is it essential to consider context while designing and developing an AV–pedestrian system? | It plays a vital role in pedestrian attitude, and its consideration will help in building a robust vehicle–pedestrian system. |
| Is it necessary to understand regional norms, social demographic variations, and adoption of traffic rules? | Understanding variations is essential, as it highly affects attitude. |
| How important is communication between pedestrians and drivers? | Pedestrian and driver actions entirely depend upon their visual communication. |
| How many people are aware of an AVs, what is their safety perception? | Both awareness and safety perception is the criteria of any technology acceptance. |
| Is it required to investigate communication mode between pedestrians and AV? | Knowing communication mode will help auto manufacturers in building a suitable communication model. |

The primary goal of this research is to understand pedestrian behavior in interactions with AVs and to investigate the factors that influence this interaction. Humans exhibit a wide range of behaviors. Age, gender, social status, local norms, and awareness all influence behavior. A survey was conducted in 27 countries between February and May 2021 with the participation of 1000 people in three languages—English, Arabic, and Urdu—to understand variation in attitudes and behaviors.

While extensive research was conducted to determine the factors that influence an individual's perception of AVs, intention to adopt AVs, and awareness of AVs, the majority of the findings have been restricted to homogeneous samples from developed countries. The contributions of our paper include an in-depth analysis of AV–pedestrian interaction in various scenarios, factors that influence this interaction, and international compassion for pedestrian behavior in various regions.

Concerning the research structure, the succeeding section depicts related work in AV awareness, safety perception, and trust in AVs. The theoretical framework and applied research methodology are explained in Sections 3 and 4, respectively. The data analysis and results are presented in Section 5, followed by a discussion of the research findings in Section 6, and finally, future work and conclusions are presented.

## 2. Related Work

The origin and study of transportation are probably as old as human beings. Before the advent of vehicles, animals were used for transportation over long and short distances. Recognizing and identifying pedestrian actions and intentions started at that time. Pedestrian studies can be divided into categories, classical pedestrian studies and pedestrian studies involving AVs [13]. Pedestrian classical studies started in the early 1950s. At that time, the focus was on the factors that influence pedestrian behavior, such as environmental factors and pedestrian factors [14]. Pedestrian studies started after the invention of the first AV [15]. In the beginning, studies focused were on AV safety perception, choices, and

attitude towards AV adoption [11], but now after successful testing and deployment of AVs in urban public transportation, researchers are focusing on pedestrian AVs interaction [16]. Currently, AVs have dedicated movement zones and specific routes because pedestrian behavior is not yet fully explored.

This section discusses related work in behavioral studies in the context of AVs. Different strategies have been adopted for this purpose; surveys and field experiments were mostly applied to understand the effect of technology on human behavior.

As mentioned earlier, initial studies focused on participant's preferences, attitudes, and choices, and data were collected by survey or by presenting hypothetical scenarios. Direshan [17], in his research, examined individuals' attitudes toward AVs by considering trust and sustainability concerns by applying the technology acceptance model (TAM). Two variables, namely perceived usefulness (PU) and perceived ease of use (PEOU), were added to existing models. Participants showed their intention to adopt the AVs based on their usefulness rather than their easiness. Kim [18] concluded that the willingness to adopt AVs and opinion about adoption depends on users' mobility profiles. It is possible to identify distinct classes of users with unique mobility profiles through Latent class cluster analysis (LCCA).

Public safety perception is considered a crucial factor in the adoption of AVs. Penmetsa's [19] study indicated that the safe interaction experience of AVs will positively affect public perception and help in the long-term adoption of AVs. Due to the different operating regulations of AVs in different regions, different public experiences were observed. Pyrialakou [20], in his research, explored the safety perceptions of the general public in terms of their potential roles as pedestrians, bicyclists, and drivers of conventional vehicles on shared roadways. It was observed that previous experience, testing, and knowledge of recent progress were positively correlated with perceived safety about AVs; on the other hand, awareness of the crashes and other safety-related incidents fueled safety concerns about AVs. Nair [21], in his research, considered public awareness of AVs and individuals' interest in using AVs, along with the perceived safety of sharing the road with AVs. The result indicated a need for adequate awareness and safety information/demonstration campaigns about AVs. One of the critical findings was the need to recognize the end-users socio-technical consideration and other human-related factors.

Pedestrian–AV communication modalities have been studied by researchers. Common modalities that were tested for this purpose include lighting patterns [22], human-like features such as moving eyes on AVs [23], and displaying messages on an LCD [24]. In one of the survey-based studies performed by P. Bazilinskyy [25], participants were asked about clarity of communication using human–machine interfaces such as images or video and patterns that automobile makers used. Participants selected textual communication as the clearest interface. In the same research, participants were questioned about the effect of color, message content, and the prospect of displaying messages from the pedestrian point of view. A display message such as 'walk' in a green font color received maximum responses, indicating an egocentric perspective by the pedestrian. B. Zandi's [26] research survey was conducted in six countries to determine which displaying messages were more apparent to pedestrians based on different scenarios. The survey result showed that messages displaying the vehicle's status are more effective than the status messages. In the second part of the survey, participants were asked about the importance of communication at different timings, including pedestrian-automated vehicle distance and the usefulness of displaying vehicle speed.

Studies in the context of AV–pedestrian behavior started a few years ago; some prominent studies are summarized in Table 2. In research [27], an international survey was conducted with 33,958 participants from 51 countries, and multilevel structural equation modeling was applied to compare the difference in perception level and adoption of AVs. Young males are observed more optimistic about current perceptions of AV. In [28], research was conducted in urban Chinese cities, where pedestrian behaviors, trust, and intention to adopt AVs were analyzed, and data were collected from government databases. One of the

important outcomes of the research was the safety concern was observed in the occluded environment. The study did not formulate behavior analysis by applying any model. In another research [29], a survey was conducted, developed, and validated to establish pedestrian receptivity towards AVs and pedestrian receptivity of fully autonomous vehicles (PBQF); 482 participants completed the survey, and a principal component analysis was performed for data analysis. Factors such as demographic effects, location, and personal innovativeness were considered. In [30], survey-based research was performed to establish perceptions, particularly with regards to the safety and acceptance of autonomous vehicles. It was observed that perceived risk varies with gender and age variation. Most of the studies performed focused on pedestrian preferences, attitudes, modalities of communication, safety perception, and trust in AVs. Very few studies on AVs– pedestrian interaction and their cross-country comparison have been made so far, and research was performed mainly in developed countries of Europe and America. According to the author's knowledge, there is not even single research conducted between developing countries of Asia and Africa together with economically developed countries in the context of AVs–pedestrian interaction.

**Table 2.** Comparison table of related work on AV–pedestrian behavioral studies.

| Reference/Study | Survey/Analysis Approach | Research Objective | Main Finding |
|---|---|---|---|
| [27] | International survey Multilevel structural equation modeling | Perceptions of AV safety Awareness of AV Cross country/cultural comparison | Young males have more optimistic and positive perceptions of AVs. |
| [28] | Nation-based survey Basic statistical analysis | Pedestrian behavior analysis Trust and intention to adopt Modality of communication | Safety concerns were observed in an occluded pedestrian environment. |
| [31] | Nation-based survey Principal Component Analysis | Trust and Intention to adopt Perceptions of AV safety Perceptions of AV safety | People who are familiar with AVs advanced assisted systems believe that AVs are more useful and safe |
| [32] | Nation-based survey Factor Analysis/ Regression Analysis | Pedestrian behavior analysis Perceptions of AV safety | Males reported a significantly higher frequency of unsafe behaviors on the road than females |
| [33] | Nation-based survey Factor Analysis | Perceptions of AV safety Trust and Intention to adopt | Pedestrians believe AV–pedestrians are less risky compared to human-operated cars |
| [30] | International survey Graphical Analysis | Trust and intention to adopt perceptions of AV safety Cross country/cultural comparison | The respondents are most concerned about crashing, malfunctioning, purchase price, liability for incidents, interaction |

## 3. Theoretical Framework

As mentioned earlier, auto manufacturers are currently targeting the commercialization of AVs globally; hence, cross-country research is required to understand the effect of social norms, demographic variations, and ethical values on pedestrian behavior. The theory of planned behavior is one of the most dominant theoretical frameworks applied to predict human behavior from the last three decades [34]. An extended version of the theory of planned behavior was applied as the base model for this purpose. The theory of planned behavior [35,36] was developed by Icek Ajzen to predict human behavior. According to this theory of planned behavior, the most influential predictor to determine someone's behavior is the individual intention to perform a behavior; intention wholly depends upon attitude, social norm, and perceived behavioral control.

The theory of planned behavior has been extensively applied in a variety of research domains, including healthcare [37–40], environmental science [41–43], supply chains [44–46],

and transportation [47–49]. For predicting pedestrian behavior, Demir [50] compared the theory of planned behavior and prototype willingness model (PWM) in pedestrian violations. Previous researchers suggested that social behavior is the dominant factor in pedestrian violation rather than planned behavior. Piazza [51] researched college students' road crossing behavior while using a mobile phone. A questionnaire-based survey was conducted to collect responses, where 4878 crossing instances were observed, and significant distraction was found by the pedestrian while using a mobile phone. In other research, Sundararajan [52] investigated pedestrian behavior while using crossing facilities. A questionnaire was used as an experimental tool, and a structural equation model (SEM) model and factor analysis were performed for data analysis. The pedestrian expectation was found as the most dominant factor. Hashemiparast's [53] research was based on road-crossing behavior in potentially risky situations; data of 562 young adults were collected using a questionnaire; 18% of participants were selected who had previous experience of a vehicle collision. Those who had previous experience of vehicle collision showed fewer safe behaviors in crossing the road than those who had not experienced an accident. Subjective norms and attitudes were also found to be influencing determinants during road crossing. In another piece of research [54], pedestrian crossing behaviors were analyzed at signalized intersections, and past behavior and travel environment were added to the model of predicting pedestrian violation. A structural equation model was used to verify the model's validity, and the research revealed that experience and context were the main influencing factors of pedestrian violation.

In general, applied TPB-based models for pedestrian behavioral analysis have limitations, including that research data were collected in specific circumstances, the small and targeted group is focused, driver–pedestrian communication is neglected, contextual factors were not given proper attention, awareness, and technology adoption was not linked to behavior prediction, and most important studies did not include different perspectives. In our proposed TPB-based AV–pedestrian interaction model, behavioral analysis was performed while considering all influencing factors.

The theory's first construct is the intention, which refers to a person's motivation to work hard to endorse the behavior and willingness to employ effort. The stronger the intention, the more likely the individual is to perform the behavior. The second construct is the attitude towards intention; attitude is one's positive or negative thoughts concerning the performance of the behavior. Behavioral beliefs and outcome evaluation influence attitude. The third construct is the subjective norm, which is how the person feels social pressure to perform the behavior. Subjective norm is predicted by normative belief and motivation to comply. Perceived behavioral control is the fourth construct and is a person's beliefs about the resources and opportunities that may facilitate or impede the behavior.

The TPB-based model for this research is depicted in Figure 2. In this proposed model, a person's intention toward any behavior is a function of three constructs: attitude, subjective norms, and behavioral control. For attitude factors such as context, pedestrian–driver visual communication, personal own psychological conditions, and habits of obeying traffic rules were all considered. For subjective norms, features such as the social norm, socio-demographic characteristics, driving habits, and awareness are selected, while perceived behavioral control is defined as affordability, safety, and appropriate communication mode.

Based on the theoretical and empirical background discussed above, five hypotheses were formed:

**Hypothesis 1 (H1).** *There is a direct relationship between attitude toward the intention and intention.*

**Hypothesis 2 (H2).** *There is an association between subjective norms and intention.*

**Hypothesis 3 (H3).** *There is an association between perceived behavioral control and intention.*

**Hypothesis 4 (H4).** *There is an association between perceived behavioral control and behavior.*

**Hypothesis 5 (H5).** *There is an association between intention and behavior.*

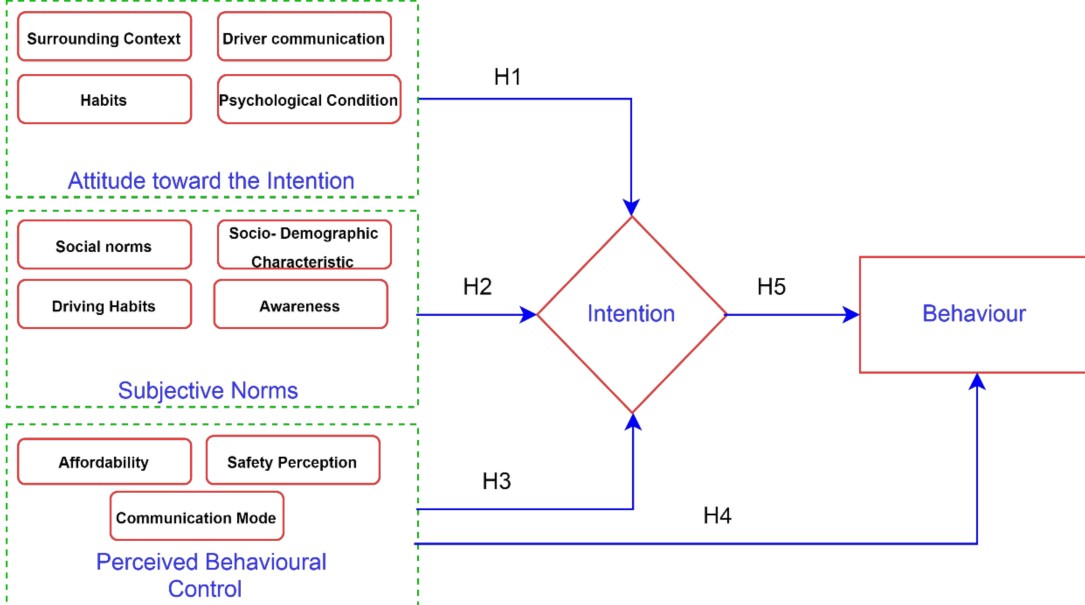

**Figure 2.** TPB-based AV–Pedestrian Interaction Model.

## 4. Methodology

### 4.1. Survey Description

A survey of 27 questionnaires was formed for analyzing pedestrian behavior in different scenarios, comprised of six items: demographic information (three questions), pedestrian behavior-based questions (six items), social norms related questions (four items), scenario-based questions (four items), driver–pedestrian behavior-based questions (four items), and people's perception towards AVs (six items). In terms of pedestrian behaviors, survey questions were distributed accordingly: attitude (11 items), out of which six items show aggressive behaviors and five items indicate positive behaviors; violations (seven items); AV interaction; and communication (six items). An average time of fifteen minutes was estimated for the survey completion.

### 4.2. Survey Timeline/Survey Phases

An online Google survey form was used as the communication mode. All survey questions were tested through the focused group of fifty participants. For the best interpretation of the question by participants, clear and specific questions were formed. After the period of four months (February 2021 to May 2021), theoretical saturation was obtained. Data editing, analyzing, and finally, the interruption was performed. The survey milestone and timeline are elaborated by the flowchart. The Figure 3 flowchart illustrates the timeline of the complete process.

## 5. Data Analysis and Results

In order to test the hypotheses of the study, SPSS 26 and SmartPLS 3 were applied. SPSS performs parametric and non-parametric statistical techniques in the context of univariate, bivariate, and multivariate analysis. SmartPLS is one of the commonly used software by researchers for investigating hypotheses using SEM analysis using the Ordinary Least Square estimation techniques [55].

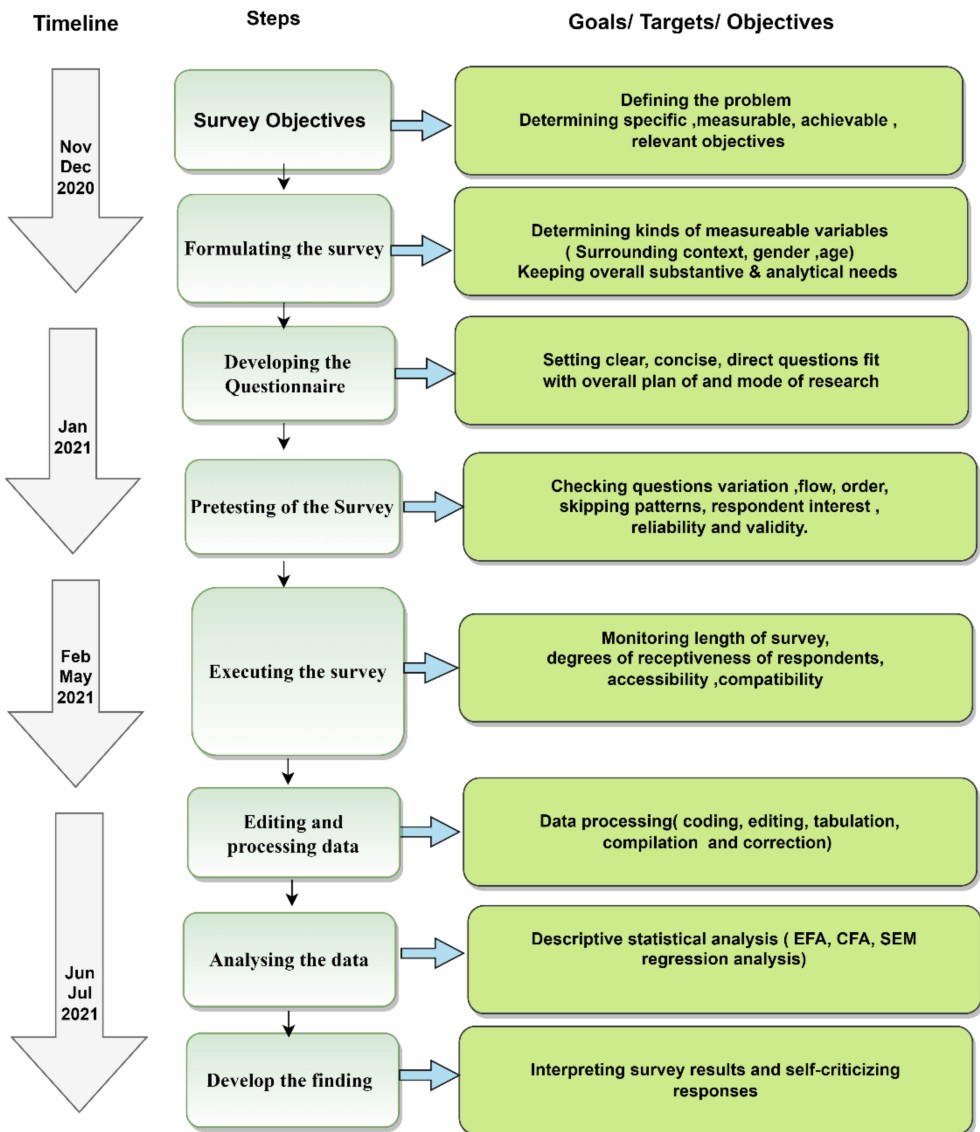

**Figure 3.** Flow Chart for the Survey Process.

Firstly, descriptive statistics (mean, standard deviation, skewness, and kurtosis) were used for demographic variables and scales. Secondly, the authors developed the scale PBQF first time; thus, exploratory factor analysis (EFA) was applied to define the factor. Exploratory Factor Analysis (EFA) is a type of multivariate statistical correlation analysis that can be used to test the validity of variables [56]. This EFA technique is used to figure out how many variables are underneath a single general variable. CFA (confirmatory factor analysis) was applied by the partial least squares structural equation modeling (PLS-SEM) method to test the model of the study. For data analysis, PLS-SEM was used to test the measures and validate the model, as well as to examine the correlations between constructs in the proposed research model [57].

Table 3 presents descriptive statistics for all the variables under consideration. In order to evaluate the symmetry of the distribution and identify missing values, a descriptive analysis was performed. All of the components' mean scores were in accordance with the normative data. Based on skewness and kurtosis results, both univariate and multivariate normality were investigated. The findings of analyzing the items and dimensions of place attachment indicated no significant deviations from normality.

**Table 3.** Descriptive Statistics.

|  | N | Minimum | Maximum | Mean | Std. Deviation | Skewness | Kurtosis |
|---|---|---|---|---|---|---|---|
| Q_01 | 965 | 1 | 5 | 4.580 | 0.959 | −2.660 | 6.555 |
| Q_02 | 965 | 1 | 5 | 4.410 | 1.093 | −1.833 | 2.372 |
| Q_03 | 965 | 1 | 5 | 4.540 | 1.008 | −2.362 | 4.696 |
| Q_04 | 965 | 1 | 5 | 4.460 | 1.129 | −2.117 | 3.337 |
| Q_05 | 965 | 1 | 5 | 4.520 | 1.037 | −2.363 | 4.695 |
| Q_06 | 965 | 1 | 5 | 4.170 | 1.130 | −1.207 | 0.566 |
| Q_07 | 965 | 1 | 5 | 4.490 | 0.921 | −1.877 | 2.992 |
| Q_08 | 965 | 1 | 5 | 4.670 | 0.780 | −2.912 | 8.830 |
| Q_09 | 965 | 1 | 5 | 4.450 | 0.949 | −1.895 | 3.222 |
| Q_10 | 965 | 1 | 5 | 4.060 | 1.226 | −1.051 | 0.054 |
| Q_11 | 965 | 1 | 5 | 3.970 | 1.252 | −0.987 | −0.107 |
| Q_12 | 965 | 1 | 5 | 4.190 | 0.966 | −1.173 | 0.783 |
| Q_13 | 965 | 1 | 5 | 4.330 | 1.024 | −1.680 | 2.248 |
| Q_14 | 965 | 1 | 5 | 4.360 | 1.004 | −1.779 | 2.650 |
| Q_15 | 965 | 1 | 5 | 4.270 | 1.094 | −1.599 | 1.719 |
| Q_16 | 965 | 1 | 5 | 4.720 | 0.705 | −3.126 | 10.702 |
| Q_17 | 965 | 1 | 5 | 4.640 | 0.749 | −2.674 | 8.074 |
| Q_18 | 965 | 1 | 5 | 4.750 | 0.567 | −3.150 | 13.528 |
| Q_19 | 965 | 1 | 5 | 4.660 | 0.728 | −2.377 | 5.650 |
| Q_20 | 965 | 1 | 5 | 3.980 | 1.098 | −0.760 | −0.166 |
| Q_21 | 965 | 1 | 5 | 3.830 | 1.101 | −0.656 | −0.248 |
| Q_22 | 965 | 1 | 5 | 3.480 | 1.371 | −0.424 | −1.085 |
| Q_23 | 965 | 1 | 5 | 3.650 | 1.324 | −0.626 | −0.737 |
| Q_24 | 965 | 1 | 5 | 3.780 | 1.396 | −0.807 | −0.672 |

*5.1. Survey Participants*

A total of nine hundred and eighty-one participants from 27 countries took part in this survey. The survey was launched in three languages, namely English, Arabic and Urdu. Nine hundred and sixty-five participants were selected for analyses; 16 were removed for incorrect answers to at least one of the check questions. The survey was created using Google Form (https://forms.gle/nachX5VfJR4MeT438 accessed on 22 December 2021). Four age groups were defined, G1 (under 18), G2 (18–40), G3 (40–60), and G4 (60+). Around 64% of participants were in age group G2. Males accounted for 75% of the sample, and females accounted for 25%. The majority of participants belonged to Asia, and their percentage was 66%. Table 4 lists the country's statistics included in our study. In order to achieve a general perception of people from all over the world, we took into account the representation of participants from different countries in our studies.

*5.2. Exploratory Factor Analysis (EFA)*

A total of 981 respondents were randomly selected from twenty-seven countries. All 24 items (questionnaire) were tested with the help of principal axis factor analysis with varimax rotation by using the SPSS 26 software. For factor analysis, Varimax rotation is applied to clarify the relationship between factors. All 24 items were met each criterion of EFA. The sample size of 965 was large enough to apply the EFA [58]. The Kaiser–Meyer–Olkin (KMO) test was used to measure the sampling adequacy of 0.842, which is considered adequate by Kaiser [59]. Bartlett's sphericity test is significant $X2(276) = 10,299.350$, which is appropriate for the factor analysis (Field [60]). Communalities values were well above 0.5. The total variance is 59.54%, which is well above 50%, as suggested by Podsakoff [61]. The first factor of the study explains the 26.63% variance, which is well below 50%. Table 5 present the final results of the pattern matrix. Factor 1 represents an attitude toward the intention, factor 2 represents subjective norms, factor 3 represents perceived behavioral control, factor 4 represents intention, and factor 5 represents pedestrian behavior.

**Table 4.** Country statistics (N = 965).

| Country | Participants % |
|---------|---------------|
| Afghanistan | 0.5 |
| Australia | 0.8 |
| Bangladesh | 0.9 |
| Belgium | 0.8 |
| Brazil | 0.5 |
| Canada | 6.4 |
| China | 7.8 |
| France | 1.5 |
| India | 4.1 |
| Indonesia | 2.7 |
| Ireland | 6.9 |
| Hong Kong | 1.1 |
| Jordan | 0.8 |
| Kuwait | 1.3 |
| Kenya | 0.7 |
| Malaysia | 10 |
| Nigeria | 2 |
| Oman | 4.6 |
| Pakistan | 14 |
| Singapore | 1.3 |
| Saudi Arabia | 12.3 |
| Sudan | 1.1 |
| Syria | 1.1 |
| Uganda | 1.65 |
| U.A.E | 1.5 |
| U.K. | 3.9 |
| U.S.A. | 9.4 |

**Table 5.** Rotated Component Matrix.

| | Component | | | | | Dimension |
|------|-------|-------|---|---|---|-----------|
| | **1** | **2** | **3** | **4** | **5** | |
| Q_01 | 0.879 | | | | | |
| Q_02 | 0.785 | | | | | Attitude Toward the Intention |
| Q_03 | 0.736 | | | | | |
| Q_04 | 0.743 | | | | | |
| Q_05 | 0.714 | | | | | |
| Q_06 | | 0.724 | | | | |
| Q_07 | | 0.503 | | | | |
| Q_08 | | 0.673 | | | | Subjective Norms |
| Q_09 | | 0.778 | | | | |
| Q_10 | | 0.792 | | | | |
| Q_11 | | 0.519 | | | | |

**Table 5.** *Cont*.

| | Component | | | | | Dimension |
|---|---|---|---|---|---|---|
| | **1** | **2** | **3** | **4** | **5** | |
| Q_12 | | | 0.561 | | | |
| Q_13 | | | 0.688 | | | Perceived |
| Q_14 | | | 0.753 | | | Behavioral Control |
| Q_15 | | | 0.820 | | | |
| Q_16 | | | | 0.715 | | |
| Q_17 | | | | 0.775 | | |
| Q_18 | | | | 0.762 | | Intention |
| Q_19 | | | | 0.597 | | |
| Q_20 | | | | | 0.689 | |
| Q_21 | | | | | 0.660 | |
| Q_22 | | | | | 0.815 | Pedestrian Behavior |
| Q_23 | | | | | 0.767 | |
| Q_24 | | | | | 0.478 | |

Extraction Method: Principal Component Analysis. Rotation Method: Varimax with Kaiser Normalization. Rotation converged in 6 iterations.

### 5.3. Confirmatory Factor Analysis

In order to apply the CFA, the current study used the partial least squares structural equation modeling (PLS-SEM). CFA was performed to ensure the results of EFA by using the SmartPLS. Various assumptions about normality and multicollinearity, as well as common method bias, were examined before going on to assess the reliability, validity, and structural model. The current study used a two-step procedure for testing the PLS-SEM [62,63].

### 5.4. Assessment of Measurement Model

Examining the measurement models is the first step in analyzing the PLS-SEM findings. The authors of [64–66] recommended that for measuring the measurement model, it is essential to test the individual item reliability/indicator loadings, internal consistency reliability, convergent validity, and discriminant validity.

Individual item reliability/indicator loadings: Indicator loadings is the first step in evaluating a measurement model. Loadings greater than 0.708 are suggested since they show that the concept explains more than 50% of the variation in the variable, as well as retain the items if loading between 0.40 and 0.70 [66]. The current study shows adequate indicator loadings; thus, individual item reliability is satisfied, as shown in Table 6.

Internal consistency reliability: is the second most important measure of reliability. Internal consistency reliability is assessed by composite reliability, Cronbach's Alpha, and rho_A. Hair [66] and Bagozzi [67] proposed a criterion of 0.7 or above for assessing composite reliability, Cronbach's Alpha, and rho_A. As shown in Table 6, the values of composite reliability, Cronbach's Alpha, and rho_A of the study's latent variables are above 0.7, indicating sufficient internal consistency reliability of the measures Hair [66].

Convergent validity: shows the extent to which a construct converges to explain the variance of its items. Fornell [68] proposes that convergent validity should be tested with average variance extracted (AVE). Furthermore, as per Chin [69], the AVE should be at least 0.50 or higher to show that a construct has good convergent validity. Referring to Table 6, every construct in this study obtained a minimum of 0.50 AVE values, which indicates that the study shows good convergent validity.

**Table 6.** Evaluation of the Measurement Model.

| Variables Name | Item Label | Factor Loading | Cronbach's Alpha | rho_A | Composite Reliability | Average Variance Extracted (AVE) |
|---|---|---|---|---|---|---|
| Attitude Toward the Intention | | | 0.849 | 0.856 | 0.892 | 0.624 |
| | Q_01 | 0.869 | | | | |
| | Q_02 | 0.806 | | | | |
| | Q_03 | 0.740 | | | | |
| | Q_04 | 0.789 | | | | |
| | Q_05 | 0.739 | | | | |
| Subjective Norms | | | 0.814 | 0.818 | 0.864 | 0.516 |
| | Q_06 | 0.702 | | | | |
| | Q_07 | 0.696 | | | | |
| | Q_08 | 0.733 | | | | |
| | Q_09 | 0.796 | | | | |
| | Q_10 | 0.744 | | | | |
| | Q_11 | 0.630 | | | | |
| Perceived Behavioral Control | | | 0.760 | 0.781 | 0.844 | 0.574 |
| | Q_12 | 0.773 | | | | |
| | Q_13 | 0.779 | | | | |
| | Q_14 | 0.728 | | | | |
| | Q_15 | 0.750 | | | | |
| Intention | | | 0.752 | 0.772 | 0.844 | 0.577 |
| | Q_16 | 0.773 | | | | |
| | Q_17 | 0.838 | | | | |
| | Q_18 | 0.777 | | | | |
| | Q_19 | 0.634 | | | | |
| Pedestrian Behavior | | | 0.807 | 0.807 | 0.866 | 0.566 |
| | Q_20 | 0.769 | | | | |
| | Q_21 | 0.758 | | | | |
| | Q_22 | 0.802 | | | | |
| | Q_23 | 0.783 | | | | |
| | Q_24 | 0.639 | | | | |

Discriminant validity: measures the degree to which a variable in the model is empirically different from other constructs. The discriminant validity was determined using the Fornell [68] criterion and the Henseler [70] HTMT ratio. As per Fornell and Larcker's [71] criteria, the square root of the AVE should be greater than the correlations among the latent variables for determining discriminant validity. HTMT values should be <0.85 or must <0.9 as per HTMT ratio [70]. Tables 7 and 8 show the results of the study, which are in the suggested ratio; therefore, it is concluded that discriminant validity is not the issue in the current study.

**Table 7.** Evaluation of the Discriminate Validity by Fornell and Larcker Criteria.

| | ATI | I | PB | PBC | SN |
|---|---|---|---|---|---|
| Attitude Toward the Intention | 0.790 | | | | |
| Intention | 0.269 | 0.759 | | | |
| Pedestrian Behavior | 0.234 | 0.372 | 0.752 | | |
| Perceived Behavioral Control | 0.284 | 0.342 | 0.441 | 0.758 | |
| Subjective Norms | 0.141 | 0.435 | 0.507 | 0.359 | 0.719 |

Note: ATI = Attitude Toward the Intention, I = Intention, PB = Pedestrian Behavior, PBC = Perceived Behavioral Control, SN = Subjective Norms.

**Table 8.** Evaluation of the Discriminate Validity by Heterotrait–Monotrait Criteria.

|  | ATI | I | PB | PBC | SN |
|---|---|---|---|---|---|
| Attitude Toward the Intention |  |  |  |  |  |
| Intention | 0.331 |  |  |  |  |
| Pedestrian Behavior | 0.287 | 0.468 |  |  |  |
| Perceived Behavioral Control | 0.333 | 0.442 | 0.519 |  |  |
| Subjective Norms | 0.161 | 0.537 | 0.627 | 0.419 |  |

Note: ATI = Attitude Toward the Intention, I = Intention, PB = Pedestrian Behavior, PBC = Perceived Behavioral Control, SN = Subjective Norms.

### 5.5. Assessment of Structural Model

As the measurement model is satisfactory, the next step is to measure the structural model in PLS-SEM. Collinearity must be checked before examining structural relationships to ensure that it does not influence the regression findings. VIF values should be <5 (Mason [71]). VIF values for both models of the current study are in the suggested range. As collinearity is not the issue, the next step is to test the hypotheses of the study. A standard bootstrapping approach with 5000 bootstraps subsamples with 965 responses was used to assess the significance of the bath coefficients [63]. Moreover, the f-square is reported, which shows the effect sizes. Values of 0.02, 0.15, and 0.35 represent the small, medium, and large $f2$ effect sizes, respectively [72].

Originally, H1 proposed that there is association between attitude toward intention and the intention. The results presented in Table 9 and Figure 4 shows that there is a statistically significant association between attitude toward the intention and intention ($\beta$ = 0.172, SE = 0.037, $t$-value = 4.713, $p$-value < 0.000, $f^2$ = 0.037, CI LL = 0.106, CI UL = 0.248). Hence, H1 is supported. Initially, H2 proposed that there is association between subjective norms and intention. Results presented in Table 9 and Figure 4 proofs that there is statistically significant association between subjective norms and intention ($\beta$ = 0.350, SE = 0.040, $t$-value = 8.725, $p$-value < 0.000, $f^2$ = 0.143, CI LL = 0.274, CI UL = 0.430). Therefore, H2 is supported. Primarily, H3 proposed that there is association between perceived behavioral control and intention. The results presented in Table 9 and Figure 4 elaborates that there is a statistically significant association between perceived behavioral control and intention ($\beta$ = 0.167, SE = 0.034, $t$-value = 4.903, $p$-value < 0.000, $f^2$ = 0.031, CI LL = 0.102, CI UL = 0.235). Consequently, H3 is supported. Originally, H4 proposed that there is association between perceived behavioral control and pedestrian behavior. The results presented in Table 9 and Figure 4 exposes that there is a statistically significant association between perceived behavioral control and pedestrian behavior ($\beta$ = 0.355, SE = 0.029, $t$-value = 12.127, $p$-value < 0.000, $f^2$ = 0.149, CI LL = 0.298, CI UL = 0.413). Hence, H4 is supported. Initially, H5 proposed that there is association between intention and pedestrian behavior. The results presented in Table 9 and Figure 4 shows that there is a statistically significant association between intention and pedestrian behavior ($\beta$ = 0.251, SE = 0.032, $t$-value = 7.910, $p$-value < 0.000, $f^2$ = 0.074, CI LL =0.188, CI UL = 0.312). Consequently, supporting H5.

**Table 9.** Testing Hypothesis Using Path Coefficients.

| H | Relationship | Std Beta | Std Error | $t$-Value | f-Square | Decision | CI LL | CI UL |
|---|---|---|---|---|---|---|---|---|
| $H_1$ | ATI -> I | 0.172 | 0.037 | 4.713 *** | 0.037 | Supported | 0.106 | 0.248 |
| $H_2$ | PBC -> I | 0.350 | 0.040 | 8.725 *** | 0.143 | Supported | 0.274 | 0.430 |
| $H_3$ | SN -> I | 0.167 | 0.034 | 4.903 *** | 0.031 | Supported | 0.102 | 0.235 |
| $H_4$ | PBC -> PB | 0.355 | 0.029 | 12.127 *** | 0.149 | Supported | 0.298 | 0.413 |
| $H_5$ | I -> PB | 0.251 | 0.032 | 7.910 *** | 0.074 | Supported | 0.188 | 0.312 |

Note: ATI = Attitude Toward the Intention, I = Intention, PB = Pedestrian Behavior, PBC = Perceived Behavioral Control, SN = Subjective Norms. *** = $p$ <0.001 (two-tailed test).

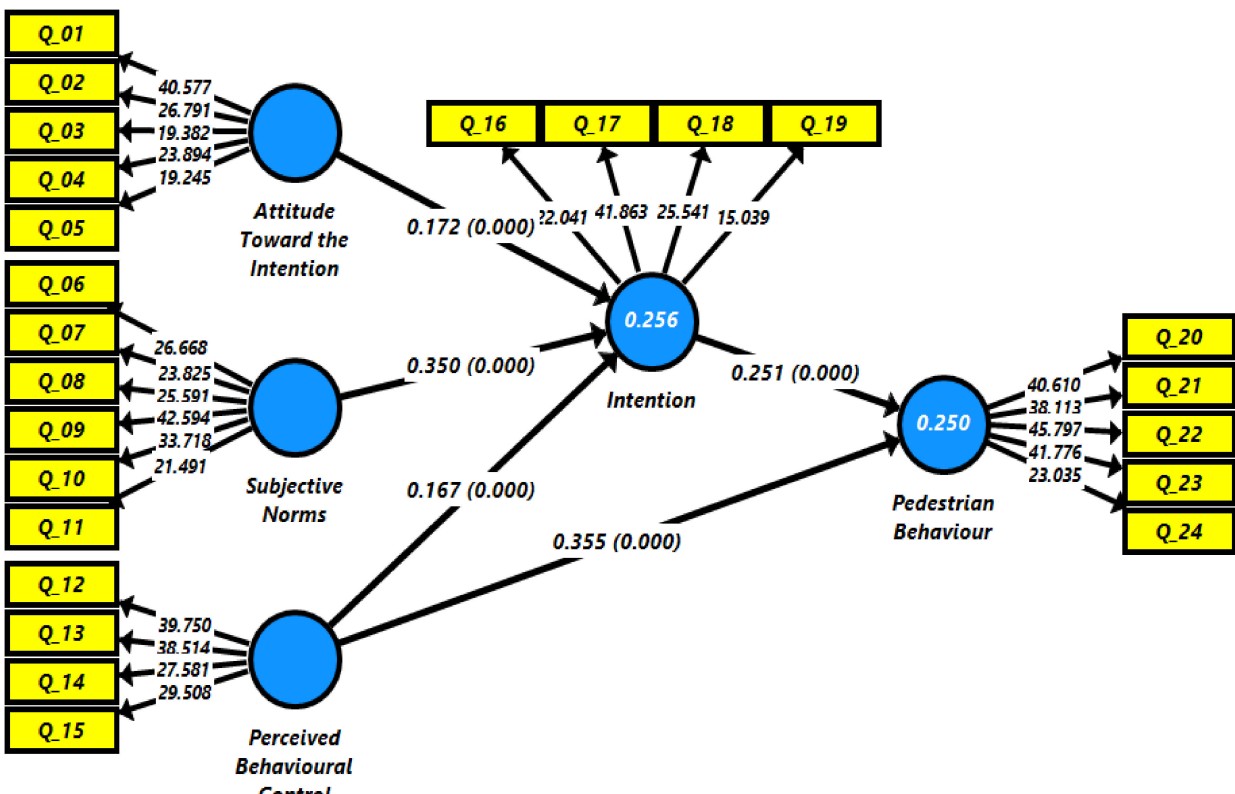

**Figure 4.** Results of the structural equation model.

### 5.6. R-Square (R2)

R- square Measure the variance that is explained in each of the endogenous variables by independent variable/s (measure the explanatory power) Shmueli [73]. In-sample predictive power is another name for the R2 Rigdon [74]. R2 values range between 0 to 1, where higher values indicate stronger explanatory power. R2 values of 0.6, 0.33, and 0.19 are regarded as substantial, moderate, and weak, respectively [63,65]. The obtained value of R2 is 0.256 for intention and 0.250 for pedestrian behavior. This shows that attitude toward the intention, subjective norms, and perceived behavioral control altogether explained 25.5% (moderate) variance in intention, and perceived behavioral control and intention explained 25% (moderate) variance in pedestrian behavior.

### 5.7. Model Predictive Relevance (Q2)

Calculating the cross-validated redundancy measure (Q2) value is another way to evaluate the prediction accuracy of the PLS path model [74]. The predictive relevance is recommended as a supplementary analysis since the goodness-of-fit (GoF) score is not adequate for model validation in PLS-SEM because it cannot distinguish between valid and invalid models [70]. According to Chin [69] and Henseler [65], a research model with a Q2 value greater than zero is considered that model has predictive relevance. The cross-validated redundancy Q2 test results are shown in Table 10. The obtained Q2 value is greater than zero, which indicates both models have predictive relevance.

**Table 10.** Construct Cross-Validated Redundancy (Q2).

|  | SSO | SSE | $Q^2$ (=1 − SSE/SSO) |
|---|---|---|---|
| Intention | 3860 | 3309.290 | 0.143 |
| Pedestrian Behaviour | 4825 | 4164.002 | 0.137 |

## 6. Discussion

The primary aim of this study was to develop and validate a pedestrian behavior survey for the full AVs. In order to find diversity in pedestrian behaviors, a TPB-based model was compared using 965 individuals from 27 countries. The designed PBQF predicted pedestrian behavior in different scenarios, specifically while crossing the road, and identified influencing factors such as context, pedestrian–driver communications, norms, psychological feelings, AV awareness, safety perception, and communication mode in pedestrian–AV interaction. For validating the usefulness of this PBQF, various approaches were applied, and the results confirmed the effectiveness of this PBQF, with a few minor modifications. In this section, the findings and outcomes in the context of pedestrian–AV behavior are explained.

### 6.1. Demographic Influences

Age is considered to be one of the significant demographic factors in behavior analysis. Both pedestrians and drivers were considered for behavior analysis. The study revealed that during driving, people are observed to be more careful when they face children and the elderly on the road. Almost 78% were vigilant while facing children and elderly people. The findings confirmed previous results [75,76], that elderly pedestrians and children are more likely to perform unintentional risky pedestrian behaviors. For driver age, no significant difference is reported in pedestrian behavior. The second factor in behavior analysis was gender, where over 40% of the pedestrian reported that they tend to be more attentive while facing female pedestrians. On the other hand, 55% reported that they treat both genders equally. Similarly, as with the driver's age, the driver's gender is not an influencing variable in this study.

In terms of comparing continents samples, European and American respondents showed less concern about gender and age variations than Asian and African respondents did. Asians showed more concern in gender and age differences in comparison with other continents.

### 6.2. Surrounding Contextual Effect and Pedestrians Responses

One of the practical challenges [47] of AVs is predicting pedestrian behavior while relating and considering their surrounding context. Responses revealed that pedestrians' response becomes uncertain if they find garbage and filth in their path while crossing the road, and can lead to any unexpected action. Only 11% agreed that they would keep focused on the road crossing and would not deviate their path. The remaining participants showed either they will change their track or decide at the time, indicating ambiguity in behavior.

The second important outcome in this context is pedestrian body action in such scenarios. Research revealed that pedestrians use different gestures when they face an indefinite situation. The majority of the participants confirmed that they would use hand indications to show their intention. Only 13% intended to use head indication, and 11% vowed that they would not perform any action.

Further insight revealed significant differences in gesture application; participants from Europe and America were more responsive than Asians and Africans.

To attain human driving perception, AVs have to reach this level of discernment, understanding full context and pedestrian gestures variations.

### 6.3. Driving Behavior

An important aspect in the development of AVs is driving behavior, which causes AVs to be considered automated vehicles and what driving style should be adopted. As indicated by studies [48–50], pedestrians built direct relationships either with driver or vehicle. Responses indicated that drivers' engagement on mobile phones or distracted driving would create high uncertainty in pedestrian action. More than 50% of responses were found to be indeterminate in such scenarios. AVs must implicitly communicate their

intent to pedestrians through driving behavior to build optimum trust between pedestrians and AVs. An AVs sudden change in speed or maneuvers by the auto-speeding or auto-braking system makes pedestrians uneasy and unsafe. Our results suggest that likeability and adaptability depend on the vehicle's driving behavior, not on who is in control. For example, sudden stopping and speeding are regarded as unintelligent and unnatural driving styles. In conclusion, AVs have to drive similar to a human for a deterministic response from pedestrians.

### 6.4. Modality of communication

Though our findings suggest that pedestrians rely on legacy behaviors (driving style) rather than the information on an external display, many participants, however, preferred additional displays on AVs.

Display text messages such as 'stop' or 'go' or other symbolic form messages need to be clear, simple, and easily readable. In pedestrian and AV interaction, the unavailability of such informal communication leads road users to feel unsafe [46]. Pedestrians have to understand and interpret display signals in a limited time. Among display options, significant diversity was observed; around one-third of the pedestrians preferred to use 'red' and 'green' lights options, while few favored lighting patterns for visual communication.

Similarly, around one-fourth found symbolic indication of 'crossing' or 'not crossing' as suitable communication. Very few believed that conventional techniques such as a horn would be suitable for communication. In summary, there was no consensus about any specific external human–machine effective AV-to-pedestrian communication.

### 6.5. AV Awareness and Safety Perception

The study paid special attention to AV awareness and safety perception. Demographic analysis of our sample across all countries revealed that younger people and males are most optimistic and receptive to AVs. It was interesting to find that older people are reluctant regarding the safety perception of AVs. This finding supports previous research [53,54] that young people have more safety awareness than other age groups. The analysis shows that young people will be early adopters and will be more frequent users of AVs than other age groups.

From a global perspective, significant variances are observed across nations. Figure 5 maps country intercept variation about AV awareness and safety perception. North American and Asian countries showed higher awareness than African countries. However, low safety perceptions of AV safety were found in Asian and African countries. Although AV development and testing are more common in North American, European, and Far East Asian countries such as China and Singapore shows more awareness of AVs and safety perception.

### 6.6. Rules Compliance

The analysis of variance revealed that people in developed countries are more responsible, whether using designated sidewalks or pedestrians waiting for longer. There were contradictory opinions of the people concerning obeying laws, and it was found that regions, religions, and local customs have a significant influence on the enforcement of the law. Therefore, it is very important to consider these factors while determining the law.

The results of the study showed that differences were observed in traffic rules compliance. Previous studies [77] also show that 74% of road traffic accidents globally occur in Asian countries due to the negligence of the people driving and walking on the road. Considering peoples' behavior alike and adopting the universal pedestrian behavior recognition model will undoubtedly lead to unavoidable negative consequences.

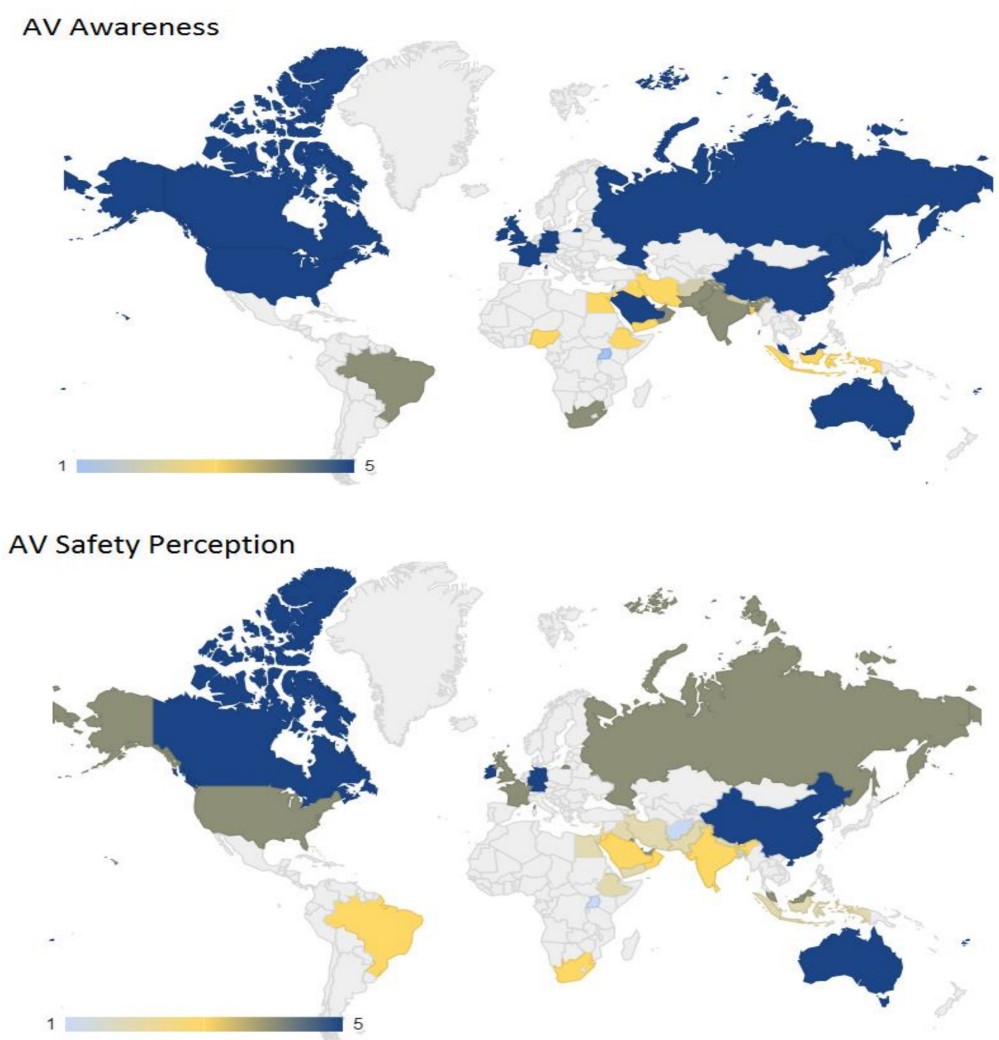

**Figure 5.** Variation in country-intercepts for AV awareness and safety perception.

In summary, the results suggest that in order to understand pedestrian behavior, it is essential to consider factors such as regional norms, social demographic differences, pedestrian context, and traffic rule adoption. The technological acceptance criterion includes both awareness and perceptions of safety. In order to enhance public perceptions of autonomous vehicles in developing nations, legal, economic, and political impediments to AV adoption must be overcome quickly. Driving style has a significant impact on pedestrian behavior. The driving style of autonomous vehicles should be identical to that of humans. The visual communication mode is recognized as the best and safest technique of communication.

## 7. Limitations and Future Work

This study proposed a TPB model for predicting pedestrian behavior in different scenarios, highlighted influencing factors that may affect pedestrian behaviors, and compared general awareness and safety perception about AVs. However, future research can improve results and overcome limitations. First, the proposed TPB model was built on theoretical assumptions for predicting behavior. Our model was built on three constructs. Behavioral prediction is a complex science [55] consisting of a sophisticated model consisting of many constructs. In future models, to improve the accuracy of prediction, more variables can be added. Second, the relationship between variables is estimated from cross-sectional data analysis. In order to gain deeper insights, other strategies such as longitudinal data analysis can be applied for further investigation. Third, pedestrian behavior classification was predicted by participants' responses, not observed behavior, while responses could

differ from the observed behavior on the road. Therefore, future research should focus on interaction with AVs to understand pedestrian behavior. Standardization of interfacing modules is also a goal that is needed to be achieved for AVs manufacturers.

## 8. Conclusions

This study was designed to validate the utility of TPB to predict pedestrian behavior around AVs considering pedestrian context, demographic values, social norms, ethical attitudes, and religiosity across different cultures in different scenarios. The designed PBQF survey was validated for 27 countries. Different statistical tests were conducted to ensure the validity of the effectiveness and validity of the proposed model. Cross-country data were collected and analyzed to explore variations and similarities in different cultures. Studies reveal that pedestrian factors such as age, gender, and socio-cultural effect should be given importance, as they may have a significant effect on pedestrian action and behavior. People's perceptions of AVs suggest that awareness and safety perceptions of AVs are important in AV–pedestrian interactions. The finding indicated that visual communication interfaces might contribute to a safer experience than conventional modalities of communication. For predicting pedestrian actions, AVs have to consider and relate all these factors for accurate prediction and intention estimation. Variations in technology awareness are also observed across different regions. External human–machine interface options are still needed to explore in order to inform pedestrians about the vehicle's current state and future actions in order to maximize safety.

**Author Contributions:** Conceptualization, F.H.; methodology, U.U.S.; software, A.A.M.; validation, S.A.-S. and A.A.M.; formal analysis, F.H.; investigation, S.A.-S.; resources, M.H.; data curation, U.U.S.; writing—original draft preparation, A.A.; writing—review and editing, U.U.S.; visualization, A.A.M. All authors have read and agreed to the published version of the manuscript.

**Funding:** This research received no external funding.

**Institutional Review Board Statement:** Not applicable.

**Informed Consent Statement:** Informed consent was obtained from all subjects involved in the study.

**Conflicts of Interest:** The authors declare no conflict of interest.

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
