# Peer review of "Application of the Theory of Planned Behavior in Autonomous Vehicle-Pedestrian Interaction"

_applsci, doi:10.3390/app12052574_

Round 1

Reviewer 1 Report

The paper describes an interesting approach and is well written and presented. Please find my comments and notes below.

It would be helpful if the authors would explain figure 1 more in detail and elaborate on how these different pedestrian model requirements were defined.

It could be interesting to read more about how the TAM model and the theory of planned behavior differentiate and how they correlate, since both make predictions about intentions. Why did the authors choose the latter one for their framework?

Isn't it logical that the CFA confirms the results of the EFA when both analyses are performed on the same data? Authors should add some references explaining that creating a model and confirming it on the same data is a valid approach.

Authors should consider to define what they mean by using "autonomous vehicles": do they refer to SAE Level 5?

3 + 4: I do not understand how a recent introduced technique can be used for three decades.

69: what is meant by this sentence?

112: maybe "participants' preferences" would be better

328-330: these sentences should be checked on correctness

Author Response

We are grateful to the anonymous reviewer for their comments and suggestions. attached are the responses to the comments of the reviewer in detail.

Reviewer 2 Report

The efforts are shown in the research. The following comments are to improve quality of the paper.

  • The abstract should focus on the problem, methodology, the main outputs, and the implications of the finding. 
  • English improvement is required, many redundant sentences and words are presented. Please revise and remove the unneeded phrases and words.
  • The introduction lacks of a story line, the novelty of the work is not well presented. I recommend for authors to include tables showing previous work works, methods, and main findings. In this table, the author can build his added value.  I see that authors jump through topics without a proper connection. Tables and figures are not well positioned. Authors should distinguish what their work and what previous scholars 
  • Page 1, introduction section, Is your research about the applications of the AVs on these fields?
  • Figure 1 is not cited in the preceding text. 
  • Page 3, after the table, the paragraph is related to the methods or results. 
  • Related work section, 
  • Studies involving AVs began after the invention of the first AV! When and where? the AV is still under development and testing.
  • Could you distinguish between pedestrian and the people/travelers in your research .
  • References are missing is some paragraphs such as first paragraph in page 4.
  • The literature section lacks a storyline. A robust conclusion based on the research gaps must be included.
  • Theoretical framework, why 3 decades, based on which references!
  • Theoretical framework, the second paragraph move it before the hypotheses 
  • The questions are not included in the paper, the google link does not work. 
  • justifications for using theses software
  • representation of the gender, age, etc. Did you make a table based on every country composition 
  • Paragraph before table 5 is unclear. 
  • Page 16, can we conclude base on this level of power (i.e., moderate). Other results need justifications where possible. 
  • The methodology did not include background on the all used statistical/mathematical methods. 
  • Justify your methods of research.

If you mean by the behavior of people to choose AV, you can include the following papers:

  • Varghese, V. and A. Jana, Impact of ICT on multitasking during travel and the value of travel time savings: Empirical evidences from Mumbai, India. Travel Behaviour, 2018. 12: p. 11-22. DOI:10.1016/j.jtrangeo.2012.02.007
  • Steck, F., et al., How autonomous driving may affect the value of travel time savings for commuting. Transportation Research Record: Journal of the Transportation Research Board, 2018. 2672(46): p. 10. DOI:10.1177/0361198118757980
  • Kouwenhoven, M. and G. de Jong, Value of travel time as a function of comfort. Journal of choice modelling, 2018. 28: p. 97-107. DOI:10.1016/j.jocm.2018.04.002
  • Coppola, P. and D. Esztergár-Kiss, Autonomous Vehicles and Future Mobility. 2019: Elsevier.
  • Berliner, R.M., et al., Travel-based multitasking: modeling the propensity to conduct activities while commuting, in Transportation Research Board 94th Annual Meeting. 2015: Washington DC, United States.
  • Malokin, A., G. Circella, and P.L. Mokhtarian, How do activities conducted while commuting influence mode choice? Using revealed preference models to inform public transportation advantage and autonomous vehicle scenarios. Transportation Research Part A: Policy and Practice, 2019. 124: p. 82-114. DOI:10.1016/j.tra.2018.12.015
  • Etzioni, S., et al., Modeling cross-national differences in automated vehicle acceptance. Sustainability, 2020. 12(22): p. 9765. DOI:10.3390/su12229765
  • Khaloei, M., A. Ranjbari, and D. Mackenzie, Analyzing the Shift in Travel Modes’ Market Shares with the Deployment of Autonomous Vehicle Technology.
  • Hamadneh, J. and D. Esztergár-Kiss. Modeling Multitasking Onboard of Privately-Used Autonomous Vehicle and Public Transport. in Scientific And Technical Conference Transport Systems Theory And Practice. 2021. Poland: Springer. DOI:10.1007/978-3-030-91156-0_7
  • Polydoropoulou, A., et al., Who is willing to share their AV? Insights about gender differences among seven countries. Sustainability, 2021. 13(9): p. 4769. DOI:10.3390/su13094769

Author Response

We are grateful to the anonymous reviewer for their comments and suggestions. Attached are our responses to the comments of the reviewer in detail.

Round 2

Reviewer 2 Report

The paper looks much better than the previous version. I added some comments that can improve the paper. The authors must put more efforst and be more careful this time. 

  • The first line of the abstract must introduce the main topic of the paper. Remove the first sentence “Fully automated vehicles (FAVs) have gained attention and proven themselves secure, 7 efficient pieces of technical revolution.”
  • Change word “developed” to “designed” in “For this purpose, pedestrian behavior questionnaire for FAVs (PBQF) is developed based on theory of 14 planned behavior (TPB)”
  • Did you distribute a survey or a questionnaire? Try to change this in the paper.
  • References are missed in some sentences like “According to a World Health Organization report, around 1.35 million people die 48 annually from traffic accidents.” Read the paper again and add the required references. 
  • Clarify in the text whether table 1 belongs to you or to previous researchers.
  • The objective of the research is different than the contribution of the work. Please write it before the objectives. Your added value to the literature.
  • In table 2, remove your work and move it to the introduction part where you can use it in defining your added value/contribution
  • Include the questions in Table 3, at least. Or add a separate table for them after survey instrument and change the heading to survey description.
  • Pedestrian studies in context of AVs started after the invention of first AV[14]. Is not clear! We do not have a complete AV to date.
  • Use one format of numbers, sixty percent or 60% in your paper
  • The language needs improvement. The answers to my comments have many typos (use software for this kind of typos and put more effort into answering) !!
  • What do you mean by surrounding context!? 
  • This result is general and obvious, please write something unique your study found. “The result shows that pedestrian surrounding environmental context, appearance, and pedestrian engagement create high uncertainty on pedestrian action.”
  • In the abstract, the result shows that pedestrian surrounding environmental context, appearance, and pedestrian engagement create high uncertainty on pedestrian action. it is unclear. I do not see a fruitful result in the abstract, based on your results, can you compare the results based on the used methods to see which factor is the most influential on the pedestrian behavior in AVs? 
  • Compare, in a paragraph, the descriptive statistics of your sample and the country’s statistics (you can include them in the appendix or just write in paragraph the main differences if any). 
  • The last sentence in the conclusion section is related to future work, please move it there. replace the sentence for whom your results will be beneficial! 
  • In the conclusion section, .... strongly influence behaviors! the behavior of whom?
  • Provide references for your justifications of my question (can we conclude based on this level of power (i.e., moderate)?).
  • Write a summary paragraph before limitations and future works to summarize your discussion or extend your conclusion. The conclusion should match the abstract and the main discussed results in the discussion.

Author Response

We are grateful to the anonymous reviewer for their comment and suggestions.  We are also providing a revised manuscript that reflects their suggestions and comments. We feel that this has resulted in a stronger manuscript.

Round 3

Reviewer 2 Report

Thanks for correcting the comments. I recommend for the authors to strength their English language. 

Table 4 include only the percentages of each country in the table and remove the others. Add paragraph explaining/commenting in that and saying that the respondents do not represent any specific population in any country. 

Define context where firstly mentioned! 

first invention of AVs! invention of AVs when it becomes available on the market. Change it to the initiation of developing AVs.

Author Response

We are grateful to the anonymous reviewer for their comment and suggestions. Below we respond to the comments of the reviewer in detail. We are also providing a revised manuscript that reflects their suggestions and comments. We feel that this has resulted in a stronger manuscript.

This manuscript is a resubmission of an earlier submission. The following is a list of the peer review reports and author responses from that submission.

Round 1

Reviewer 1 Report

- It is recommended to include the contribution of the research at the end of the abstract at line 24.

- The source of Fugure1 should be mentioned in line 67.

- Missing question marks in Table1 in research question section.

- 1. Introduction: Page 1: lines 38-40. Some resources that may be helpful for the authors: https://doi.org/10.1080/15568318.2020.1798571 https://doi.org/10.3390/joitmc6040106 https://doi.org/10.1016/j.trc.2018.12.003

Adding these reference would be helpful with regards to challenges needed to overcome for AVs.

- There are some mistakes on reference citations within the text, e.g., Bazilinskyy in line 161, and also Zandi in line 167.

- 2. Related Work: It is recommended to add more related work which has been done recently.

- It would be more helpful if the authors mention the source of information provided in lines: 195-203

- 5. Results:

Exploratory factor analysis (EFA) - line 308

It would be helpful for the readers if the authors provide more detailed results of EFA analysis, e.g., presenting rotated component matrix.

Confirmatory factor analysis – line 320

It would be helpful for the readers if the authors provide more detailed results of CFA analysis.

- The discussion part is well developed, however, it is suggested to improve the quality of the conclusion section.

- Please check language, punctuation, and grammar throughout the entire manuscript with the help of a native speaker (highly recommended). There are many small mistakes (e.g., grammatical issues within lines 168 and line 178. I would advise the authors to seek help within the language correction of the paper. Or typos issue in line 206: missing full stop at the end).

Author Response

Application of the Theory of Planned Behavior in Autonomous
Vehicle-Pedestrian Interaction.

We are grateful to the anonymous reviewers for their comments and suggestions. Below we respond to the comments of each reviewer in detail. We are also providing a revised manuscript that reflects their suggestions and comments. We feel that this has resulted in a stronger manuscript.

Response to Reviewer-1

( All changes made are highlighted by  purple colour,where required)

Comment -1

It is recommended to include the contribution of the research atthe end of the abstract at line 24.

Authors’ Response/Clarification

Thank you for highlighting this important point, contribution of this research study has been added in abstract.

Comment -2

The source of Fugure1 should be mentioned in line 67.

Authors’ Response/Clarification

Thank you for highlighting this important point, actually this figure is author's own drawing.

Comment -3

Missing question marks in Table1 in research question section.

Authors’ Response/Clarification

Thank you for highlighting this , question marks in Table1 has been added

Comment -4 home

Introduction: Page 1: lines 38-40. Some resources that may behelpful for the authors:

https://doi.org/10.1080/15568318.2020.1798571

https://doi.org/10.3390/joitmc6040106

https://doi.org/10.1016/j.trc.2018.12.003

Adding these reference would be helpful with regards to challenges needed to overcome for AVs.

Authors’ Response/Clarification

We gladly take this suggestion and found these refrences relevant to the AVs, refrences has been added and used in intrtoduction  section.

Comment -5

Introduction: There are some mistakes on reference citations within the text,e.g., Bazilinskyy in line 161, and also Zandi in line 167.

Authors’ Response/Clarification

Thank you for poiting about correction. The mentioned mistake has been rectified in the revised version of the manuscript as suggested.

Comment -6 home

Related Work: It is recommended to add more related work which has been done recently.

Authors’ Response/Clarification

We gladly take this important suggestion and have added a few of the very recently published literature relevant to the studied subject in order to further strengthen our research work as suggested.

Comment -7 home

Related Work: It would be more helpful if the authors mention the source of information provided in lines: 195-203

Authors’ Response/Clarification

Thank you for highlighting this important point, the sorce has been added in the manuscript.

Comment -8

Results:

Exploratory factor analysis (EFA) - line 308

It would be helpful for the readers if the authors provide more detailed results of EFA analysis, e.g., presenting rotated component matrix.

Confirmatory factor analysis – line 320

It would be helpful for the readers if the authors provide more detailed results of CFA analysis.

Authors’ Response/Clarification

We gladly take this important suggestion and have added rotated component matrix in Exploratory factor analysis (EFA) and have added details in both sections. More descption about CFA analysis has been added.  

 Comment -9

The discussion part is well developed, however, it is suggested to improve the quality of the conclusion section.

Authors’ Response/Clarification

Thanks for your comliments , we gladly take this important suggestion and have improved overall quality. 

Comment -10

Please check language, punctuation, and grammar throughout the entire manuscript with the help of a native speaker (highlyrecommended). There are many small mistakes (e.g., grammatical issues within lines 168 and line 178. I would advise the authors toseek help within the language correction of the paper. Or typosissue in line 206: missing full stop at the end).

Authors’ Response/Clarification

We gladly take this important suggestion and for improving grammatical issues and other mistakes, complete manuscript has been proof read by Native speaker.

Reviewer 2 Report

Overall, the authors provide a good analysis of the utility of TPB to predict autonomous vehicle–pedestrian behavior. Figure 3 especially provides very good insight into the goals and objectives of the research. 

Some comments to improve and correct errors are listed below.

General comments:

-Note that the abstract, main text, and figure/table/scheme captions are treated separately for abbreviations. This means that you need to define the abbreviation the first time you use it in each part.

-When defining abbreviations, either choose to use small letters or capital letters (e.g., line 128 for LCAA you used capital letters, while PU abbreviation in line 124 is in small letters)

-All figures look blurry; note that the recommended figure resolution is a minimum of 600 dpi

-Figures 3 and 4, and table 6 are not mentioned anywhere in the text

-It would be useful to add the graphical representation of the flowchart for the used methodology for data analysis

Thorough spellcheck needed, e.g.:
-line 414 participants from European and American
-line 104 1950 -> 1950s
-line 139 AVS -> AVs
-line 116 spacing after punctuation missing
-line 109-114 very long sentence, hard to read
-line 76 missing punctuation
-some abbreviations defined multiple times

Introduction:
-When a vehicle is on the way to the road, it needs to understand everything for safety and smooth driving accurately -unclear meaning, rephrase the sentence
-line 65 figure name is already given below the figure
-line 69-76 what is mean model? please elaborate

Related work:
-every section starts with: in some surveys

Results:
-Abbreviations RMSEA, TLI, CFI, and GFI should be defined the other way around, full name in text and abbreviations in brackets
Descriptive statistics, correlation and reliability subsection
-unify the spacing when stating values of mean, S.D, and r
-line 358 - why is beta written as a word when you used a greek letter earlier?

Discussion:
AV's Awareness and Safety Perception subsection
-line 456,457 Together with this, it indicates young people will be early adopters, likely to be frequent AVs - unclear meaning

Author Response

Application of the Theory of Planned Behavior in Autonomous
Vehicle-Pedestrian Interaction.

We are grateful to the anonymous reviewers for their comments and suggestions. Below we respond to the comments of each reviewer in detail. We are also providing a revised manuscript that reflects their suggestions and comments. We feel that this has resulted in a stronger manuscript.

Response to Reviewer-2

Comment -1

Note that the abstract, main text, and figure/table/scheme captions are treated separately for abbreviations. This means that you need to define the abbreviation the first time you use it in each part.

Authors’ Response/Clarification

Thank you for highlighting these important points, all the and other mistakes including captions have been thoroughly checked in the revised manuscript.

Comment -2

When defining abbreviations, either choose to use small letters or capital letters (e.g., line 128 for LCAA you used capital letters,while PU abbreviation in line 124 is in small letters)

Authors’ Response/Clarification

Thank you for highlighting this important point, all abrevations has been checked and corrected  carefully.

Comment -3

All figures look blurry; note that the recommended figure resolution is a minimum of 600 dpi

Authors’ Response/Clarification

Thank you for highlighting this important point, all figures has redrawn as per 600 dpi  

Comment -4

Figures 3 and 4, and table 6 are not mentioned anywhere in thetext

Authors’ Response/Clarification

We gladly take this correction and figures and tavle has been refrened in text.

Comment -5

It would be useful to add the graphical representation of theflowchart for the used methodology for data analysis

Authors’ Response/Clarification

Thank you for this important and wothy suggestion , Flow chart has been created and added in revised version.

Comment -6

Thorough spellcheck needed, e.g.

line 414 participants from European and American-

line 104 1950 – 1950s-

line 139 AVS - AVs-

line 116 spacing after punctuation missing-

line 109-114 very long sentence, hard to read-

line 76 missing punctuation-some abbreviations defined multiple times

Authors’ Response/Clarification

Thank you for highlighting these mistakes. In revised manuscript all these mistakes has been rectified.

Comment -7

Introduction:-When a vehicle is on the way to the road, it needs to understand everything for safety and smooth driving accurately -unclearmeaning,

rephrase the sentence

-line 65 figure name is already given below the figure

-line 69-76 what is mean model? please elaborate)

Authors’ Response/Clarification

Thank you for highlighting these important points, all these corrections have been made in revised manuscript.

Comment -8

Related work: Every section starts with: in some surveys

Authors’ Response/Clarification

Thank you for highlighting this important point, senetenes has been rephrased in revised manuscript

 Comment -9

Results: Abbreviations RMSEA, TLI, CFI, and GFI should be defined the other way around, full name in text and abbreviations in brackets . Descriptive statistics, correlation and reliability subsection-unify the spacing when stating values of mean, S.D, and r-line 358 - why is beta written as a word when you used a greekletter earlier?

Authors’ Response/Clarification

Thank you for this correction. The mentioned mistake has been rectified in the revised version of the manuscript as suggested.

Comment -10

Discussion: AV's Awareness and Safety Perception subsection-line 456,457 Together with this, it indicates young people will beearly adopters, likely to be frequent AVs - unclear meaning

Authors’ Response/Clarification

We gladly take this important suggestion complete senetence  has been repahrased and restructured to give reader clear menaining.

Reviewer 3 Report

Overall, I am convinced that the authors put a lot of effort in conducting and reporting their research. The topic is timely and it is great having data from non-western countries.

However, the quality of the research is in my opinion not sufficient yet to be published in a reviewed scientific journal. There is miscinception of the Theory of Planned Behavior - in my opinion it is not suitable theoretical framework to address the behavior that the authors are looking at, i.e. interaction between pedestrians and AVs. The scale used do not align with the approach of psychometric measurement of TPB an also the results are misinterpreted. I cannot recognize good differentiation between the measured psychological constructs, in fact one can see way too high correlation between the scale so the validation of a instrument is not justified. Also, TPB has already well developed scale which can be adapted and expand but the used instrument has unfortunately nothing to do with it.

I suggest the authors to rethink whether they want to use TBP as a framework or rather try to make something different out of their research and their data by analyzing first more descriptive the results and derive more simple recommendation for car manufacturers and AV development rather than trying to press it in this framework.

I hope that the authors will accept these comments as constructive as they are meant to be. 

Reviewer 4 Report

This is wrong in every aspect possible: Methods, analyses, reporting, the structure of the paper. Has this been "written" by some AI?

Round 2

Reviewer 2 Report

The authors addressed all the comments provided in the first review. The subject of the article is topical and important from the point of pedestrian behavior and awareness. 
Introduction: The introduction is of sufficient scope and not just general.
Abstract: The abstract is of good quality, arouses interest, consists of the goals, and the ability to use the results.
Methodology: The description of the method used and the presentation of the results are clear and comprehensible. 
Results: The presentation of the results is sufficiently detailed.
Conclusion: The conclusions part appropriately summarizes the previous chapters and identifies further research opportunities.

The paper has a logical route and is understandable. The article may be published as-is.